# A computational method for detection of ligand-binding proteins from dose range thermal proteome profiles

Nils Kurzawa [1,2,5], Isabelle Becher [1,5], Sindhuja Sridharan [1,3], Holger Franken[3], André Mateus [1], Simon Anders [4], Marcus Bantscheff [3✉], Wolfgang Huber [1✉] & Mikhail M. Savitski [1✉]

Detecting ligand-protein interactions in living cells is a fundamental challenge in molecular biology and drug research. Proteome-wide profiling of thermal stability as a function of ligand concentration promises to tackle this challenge. However, current data analysis strategies use preset thresholds that can lead to suboptimal sensitivity/specificity tradeoffs and limited comparability across datasets. Here, we present a method based on statistical hypothesis testing on curves, which provides control of the false discovery rate. We apply it to several datasets probing epigenetic drugs and a metabolite. This leads us to detect off-target drug engagement, including the finding that the HDAC8 inhibitor PCI-34051 and its analog BRD-3811 bind to and inhibit leucine aminopeptidase 3. An implementation is available as an R package from Bioconductor (https://bioconductor.org/packages/TPP2D). We hope that our method will facilitate prioritizing targets from thermal profiling experiments.

[1] European Molecular Biology Laboratory, Genome Biology Unit, Meyerhofstrasse 1, Heidelberg 69117, Germany. [2] Faculty of Biosciences, Heidelberg University, Heidelberg 69120, Germany. [3] Cellzome GmbH, GlaxoSmithKline, Meyerhofstrasse 1, Heidelberg 69117, Germany. [4] Center for Molecular Biology of Heidelberg University (ZMBH), Im Neuenheimer Feld 282, Heidelberg 69120, Germany. [5]These authors contributed equally: Nils Kurzawa, Isabelle Becher. ✉email: marcus.x.bantscheff@gsk.com; wolfgang.huber@embl.org; mikhail.savitski@embl.de

Studying ligand–protein interactions is essential for understanding drug mechanisms of action and adverse effects[1,2], and more generally for gaining insights into molecular biology by monitoring of metabolite– and protein–protein interactions[3–10]. Thermal proteome profiling (TPP)[1,11,12] combines quantitative, multiplexed mass spectrometry (MS)[13] with the cellular thermal shift assay[14] and enables proteome-wide measurements of thermal stability, by quantifying non-denatured fractions of cellular proteins as a function of temperature. TPP has been used to study binding of ligands and their downstream effects in cultured human[1,2] and bacterial cells[15,16], and has recently been adapted to animal tissues and human blood[17].

In addition to temperature, non-denatured fractions of cellular proteins can be measured as a function of other variables, such as ligand concentration. In the two-dimensional (2D)-TPP experimental design, both temperature and ligand concentration are systematically varied[2]. In comparison to the original proposal for TPP that only varied a temperature range (TPP-TR), this method overcomes the problem that different proteins may be susceptible to stability modulation at different compound concentrations or temperatures. Thus, 2D-TPP can greatly increase sensitivity and coverage of the amenable proteome. However, while statistical analysis for the TPP-TR assay is well established[11,18], similar approaches for 2D-TPP have been hampered by its more complicated experimental

design. 2D-TPP employs a multiplexed MS analysis of samples in the presence of $n$ ligand concentrations (including a vehicle control) at $m$ temperatures (Fig. 1a). Thus, for each protein $i$, a $m \times n$ data matrix $Y_i$ of summarized reporter ion intensities is obtained. However, these matrices contain non-randomly missing values, usually at higher temperatures, due to differential thermal stability across the proteome, i.e., some proteins may fully denature at some of the temperatures used in the experiment and thus will not be quantified at these temperatures.

In the approach of Becher et al.[2], nonlinear dose–response curves were fitted to each protein for each individual temperature. Subsequently, hits were defined by applying bespoke rules, including a requirement for two dose–response curves at consecutive temperatures to both have $R^2 > 0.8$ and a fold change of at least 1.5 at the highest treatment concentration. However, this approach, with its reliance on preset thresholds, has uncontrolled specificity (e.g., there is no explicit control of the false discovery rate (FDR)) and, as a consequence, may have suboptimal sensitivity if, e.g., the thresholds are too stringent.

In this work, we present a statistical method for FDR-controlled analysis of 2D-TPP data. By benchmarking our approach on a synthetic dataset, we demonstrate that the approach controls the FDR. Application of the approach to previously published and newly acquired 2D-TPP datasets of epigenetic drugs showcases the

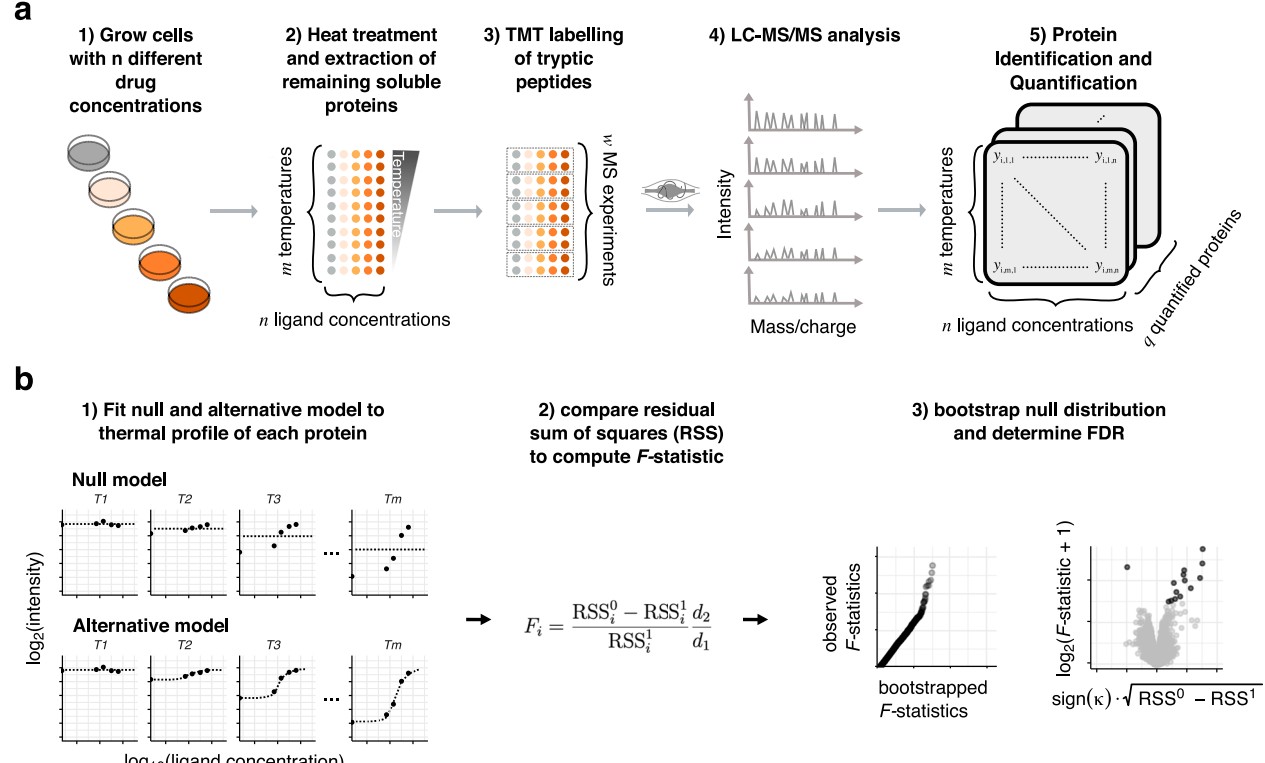

**Fig. 1 Illustration of the 2D-TPP experimental setup and our computational analysis approach. a** 2D-TPP protocol: Cells are grown in the presence of $n$ different concentrations of a ligand of interest. Each sample is divided into $m$ aliquots, each of which is subjected to one of $m$ temperatures, and the remaining soluble proteins are extracted. Proteins are digested with trypsin and labeled with TMT, such that one set of TMT labels is used for all concentrations and two adjacent temperatures. $w = m/2$ MS runs are performed, peptides are identified by database search and quantified signal is aggregated at the protein level. **b** Illustration of fitted curves under the null and alternative model, and how obtained residuals are used to find proteins significantly altered in thermal stability—and thus potential ligand interactors—via an $F$-statistic (example fits for the null and alternative model are shown in Supplementary Fig. 1). The $q$–$q$ plot on the right compares bootstrapped and observed $F$-statistics. Although the majority of quantiles of the two distributions align, the top observed $F$-statistics, corresponding to the true positives in the dataset, are shifted off-diagonal. The results can be represented as volcano plots, highlighting significant hits. RSS: residual sum of squares; $\text{sign}(\kappa) \times \sqrt{\text{RSS}^0 - \text{RSS}^1}$: measure of effect size—how much more variance is explained by the alternative model compared to the null—and direction, i.e., a positive sign for stabilized proteins, negative for destabilized ones; $\log_2(F\text{-statistic} + 1)$: the transformation is used for visualization purposes only, the addition of 1 guarantees that logarithm-transformed values remain bounded as $F$ approaches 0.

discovery of novel off-targets, including leucine aminopeptidase 3 (LAP3) for the compounds PCI-34051 and BRD-3811. We provide an open-source software implementation of our method (https://bioconductor.org/packages/TPP2D).

## Results

**Design of models for ligand dose range thermal profiles.** We developed an approach that fits two nested models to protein abundances obtained from 2D-TPP. Our approach adapts and extends a method by Storey et al.[19] for the analysis of microarray time-course experiments. The null model allows the soluble protein fraction to depend on temperature, but not concentration, as expected for proteins with no treatment-induced change in thermal stability. The alternative model fits the soluble protein fraction as a sigmoid dose–response function of concentration, separately for each temperature. This choice of the alternative model can be justified biophysically, as with increasing ligand dose, a higher fraction of a protein's population will be amenable for stabilization[14]. To increase estimation precision, the model's degrees of freedom are reduced by sharing or constraining certain function parameters across temperatures: for each protein, slope and direction of the response (destabilization or stabilization) are set to be the same across temperatures, and the inflection point, i.e., the half-maximal effective concentration in $-\log_{10}$ space (pEC50), is required to increase or decrease linearly with temperature (Supplementary Fig. 1). The residual sums of squares of the two models are compared to obtain, for each protein, an $F$-statistic. In addition, we implemented an optional empirical Bayes moderation[20] of these $F$-statistics by shrinking the denominator towards the average value among all proteins with similar number of observations. This statistic has no analytically known null distribution, so we calibrate it with an adaptation of the bootstrap approach of Storey et al.[19]. Briefly, residuals from the alternative model are resampled and added back to the null model estimate to simulate the case where there is no concentration effect of the

ligand. This resampling scheme takes into account the noise dependence of measurements within individual MS runs. Overall, our approach allows the detection of ligand–protein interactions from thermal profiles (DLPTP; Fig. 1b) and is implemented as a package for the statistical environment and language R (https://bioconductor.org/packages/TPP2D).

**Benchmarking DLPTP on a synthetic dataset.** To evaluate whether our method implementation controlled FDR as expected and how its sensitivity compared to the threshold-based approach of Becher et al.[2], we created a synthetic dataset. This dataset was composed of 5000 simulated protein thermal profiles expected under the null hypothesis of no ligand effect, with independent Gaussian noise with standard deviations observed for real datasets. In addition, 80 protein profiles known to be true positives were obtained from various datasets and spiked in. We applied our method to this dataset using 100 rounds of bootstrapping and compared nominal FDR to observed FDR (Fig. 2a). Although we found that the standard version of DLPTP was well calibrated in terms of FDR, the moderated version was conservative (Fig. 2a). However, it was observed that the moderated variant of DLPTP, which is the default in the software and was used for all further analyses in this manuscript, showed a better sensitivity-specificity tradeoff than the standard approach and the bespoke thresholds (Fig. 2b).

**Re-analysis of previously published datasets using DLPTP.** We applied our approach by re-analyzing previously published 2D-TPP datasets. For the pan-HDAC inhibitor panobinostat[21] profiled in intact HepG2 cells[2], we recovered all previously reported on- and off-targets of the drug based on this dataset[2]: HDAC1, HDAC2, TTC38, PAH, FADS1, and FADS2 (Fig. 3a, Supplementary Figs. 2a, c and 3a, b, and Supplementary Data 1), except HDAC6, which showed a noisy, non-sigmoidal profile in this dataset (Supplementary Fig. 3e and Supplementary Data 1). In

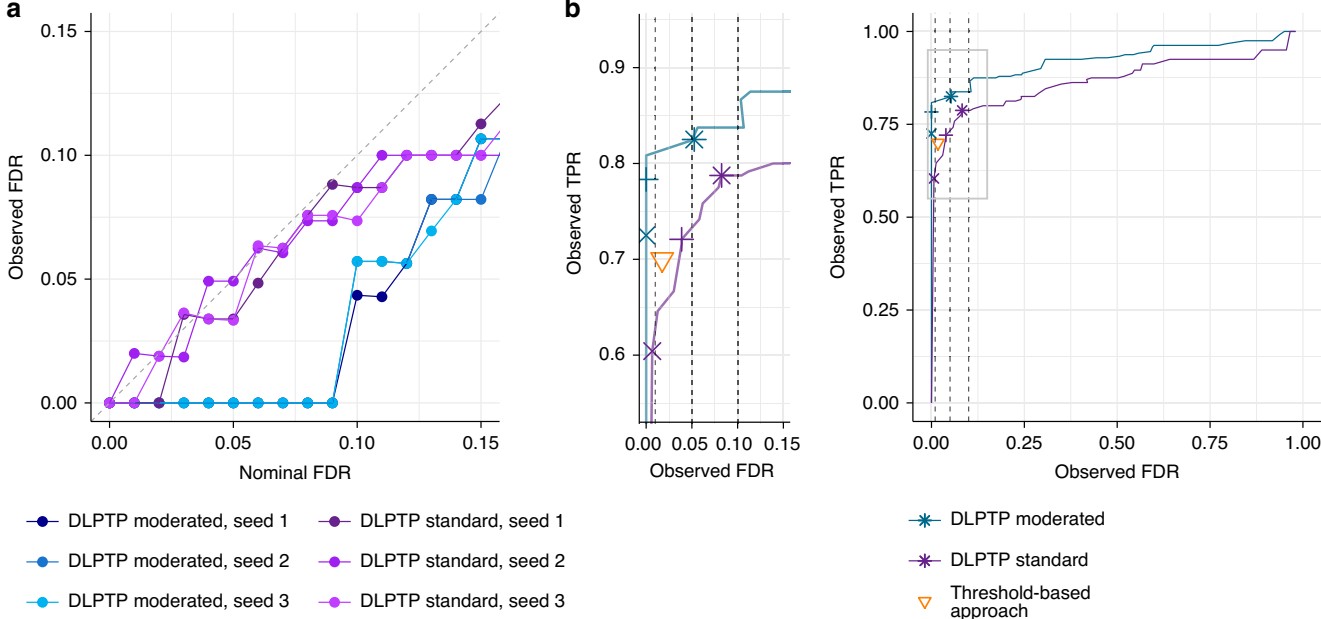

**Fig. 2 Benchmarking of DLPTP on a synthetic dataset confirms FDR control and high sensitivity. a** Observed FDR versus nominal FDR for 3 random seeds for bootstrapping ($B = 100$) with the standard and moderated DLPTP method applied to a synthetic dataset. **b** Observed TPR versus observed FDR curves for the standard and moderated DLPTP method and the threshold-based approach. Shown is the average of three different random seeds for both DLPTP methods with 100 bootstraps. Vertical lines correspond to observed FDR at 1, 5, and 10%, $x$ represents results obtained with different DLPTP versions at 1%, + at 5%, and * at 10% nominal FDR. FDR: false discovery rate; TPR: true positive rate.

addition, we detected two zinc-finger transcription factors, ZNF148 and ZNF384, and the oxidoreductase DHRS1 to significantly stabilize upon panobinostat treatment (Supplementary Fig. 3c, d and Supplementary Data 1). These observations are in line with a recent report stating that panobinostat can bind zinc-finger transcription factors and that products arising through metabolism of the drug can stabilize DHRS1[17].

Next, we re-analyzed a dataset probing the BET bromodomain inhibitor JQ1[22] in THP1 cell lysate[23]. We recovered all previously reported targets: BRD2, 3, 4, and HADHA, an enzyme with acetyltransferase activity (Fig. 3b, Supplementary Fig. 2b, d, and Supplementary Data 1). These analyses showcase that DLPTP can

be applied to 2D-TPP experiments acquired in intact cells as well as in lysates.

**Thermal profiling of the HDAC8 inhibitor PCI-34051 reveals LAP3 as a potent off-target.** Next, we performed a 2D-TPP experiment in HL-60 cells with the epigenetic inhibitor PCI-34051 (Fig. 4a), a compound reported to selectively inhibit HDAC8 that was suggested as a potential treatment for multiple types of T-cell leukemia[24]. Analysis of the dataset with DLPTP revealed 154 proteins significantly changing in thermal stability which enriched for the biological processes—oxidation–reduction process and

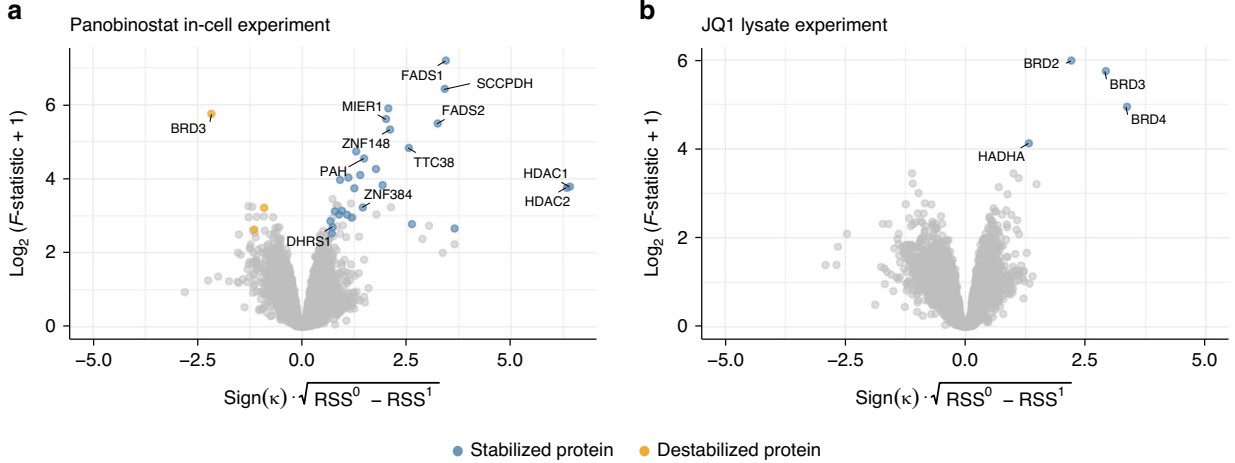

**Fig. 3 DLPTP recovers known drug–protein interactions from published datasets. a** Volcano plot for the 2D-TPP dataset acquired upon treatment of HepG2 cells with the HDAC inhibitor panobinostat[2]. **b** Analogous to **a**, for the JQ1 2D-TPP dataset acquired in THP1 lysate[23]. **a**, **b** Blue points represent proteins that were detected as stabilized by the drug treatment and orange as destabilized, at 10% FDR. Axes are outlined in Fig. 1b.

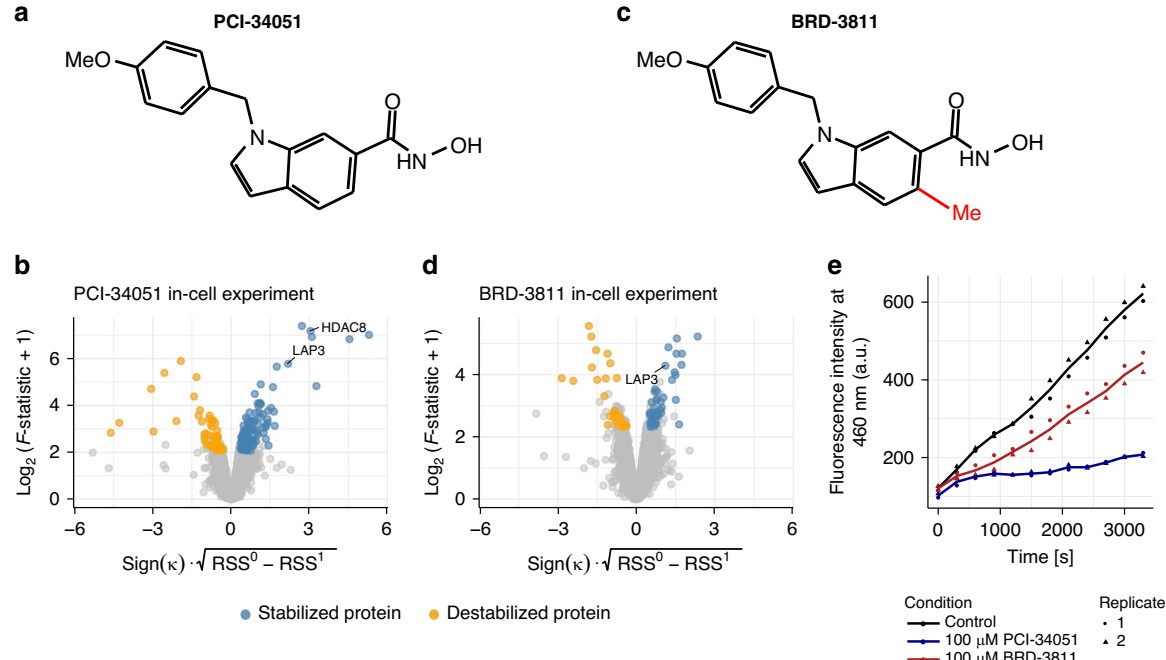

**Fig. 4 DLPTP reveals LAP3 as a target of PCI-34051 and BRD-3811. a** Chemical structure of the HDAC8 inhibitor PCI-34051. **b** Volcano plot of the 2D-TPP experiment with PCI-34051 in HL-60 cells. **c** Chemical structure of BRD-3811, an analog of PCI-34051 in which the Zn2+ chelating hydroxamic acid (HA) group is sterically hindered by a methyl group (highlighted in red). **d** Volcano plot of the 2D-TPP experiment with BRD-3811 in HL-60 cells. **e** Fluorescence intensity measured over time in a fluorometric leucine aminopeptidase assay with recombinant LAP3 in the presence of PCI-34051, BRD-3811, or vehicle control. Axes in **b** and **d** are outlined in Fig. 1b.

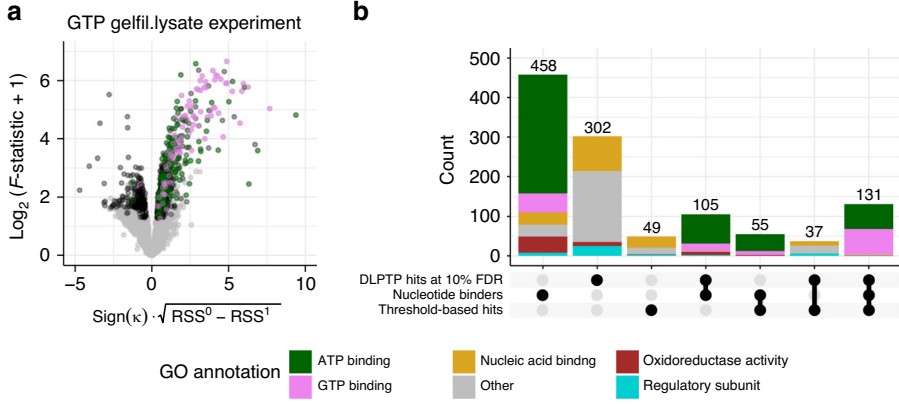

**Fig. 5 DLPTP recovers annotated GTP and ATP interactors from a 2D-TPP experiment profiling GTP–protein interactions. a** Volcano plot for the analysis of the GTP gel-filtered lysate dataset. **b** Upset plot of set intersections of annotated nucleotide-binding proteins, stabilized hits found with the threshold-based approach, and stabilized hits found by DLPTP. Axes in **a** are outlined in Fig. 1b.

carboxylic acid metabolic process (hypergeometric test, odds ratio: 3.5, adjusted $p = 5 \times 10^{-11}$ and odds ratio: 2.8, adjusted $p = 7 \times 10^{-6}$, respectively), likely reflecting the cellular response to the drug treatment and not direct drug targets. Apart from proteins reflecting these gene sets, we found the reported target HDAC8 ($pEC_{50,PCI-34051} = 6.4$) and LAP3 ($pEC_{50,PCI-34051} = 5.9$) among the top stabilized hits (Fig. 4b and Supplementary Fig. 4a). The expression of LAP3 correlates with hepatocellular carcinoma cell proliferation[25] and its inhibition suppresses invasion of ovarian cancer[26]. To follow-up our identification of LAP3, as a potential off-target of PCI-34051, we turned to an analog of the drug: BRD-3811 (Fig. 4c), in which the $Zn^{2+}$ chelating hydroxamic acid (HA) group is sterically hindered from binding HDAC8 by an additional methyl group. A 2D-TPP experiment in HL-60 cells with BRD-3811 showed no significant stabilization of HDAC8, as expected. However, we again found LAP3 as a significant target (Fig. 4d and Supplementary Fig. 4b). To investigate whether LAP3 function was inhibited by binding of either compound, we performed an in vitro fluorometric leucine aminopeptidase assay using a recombinant LAP3 enzyme. Both compounds inhibited recombinant LAP3 peptidase activity (Fig. 4e). However, BRD-3811 showed a smaller effect, in line with a 10-fold lower EC50 measured in the 2D-TPP experiment ($pEC_{50,BRD-3811} = 5.0$), which suggests that the binding of both molecules to LAP3 might be mediated via the HA group, but dampened by the additional methyl group in the case of BRD-3811.

In conclusion, 2D-TPP of the analog compounds PCI-3405 and BRD-3811 and DLPTP analysis revealed their intracellular target space (Supplementary Data 2) and showed that both bind and inhibit LAP3, a potentially interesting ovarian cancer and hepatocellular carcinoma target.

**DLPTP recovers GTP- and ATP binders from a 2D-TPP experiment profiling GTP.** So far, we focused on the application of DLPTP to detect drug–protein interactions. To evaluate whether our approach generalized to 2D-TPP datasets profiling small molecules comprising a broad range of affinities to their target proteins, such as metabolites, we performed a 2D-TPP experiment in gel-filtered lysate of Jurkat cells treated with a concentration range from 0 to 0.5 mM of NaGTP (guanosine 5'-triphosphate sodium salt). The gel filtration leads to a depletion of endogenous metabolites and thus makes metabolite-interacting proteins, which often otherwise remain bound to metabolites in lysates, particularly susceptible to bind to externally supplied metabolites. We applied DLPTP to the acquired dataset and called hits at 10% FDR. Among the significantly stabilized proteins, proteins annotated for the Gene

Ontology terms "GTP binding" (hypergeometric test $p < 2.2 \times 10^{-16}$, odds ratio: 4.6) and "ATP binding" (hypergeometric test $p = 10^{-12}$, odds ratio: 2.2) were significantly enriched (Fig. 5a, Supplementary Fig. 5a, and Supplementary Data 3). This finding is in line with recent reports showing that many ATP binders may bind to both ATP and GTP[5,6]. In comparison to a threshold-based approach, DLPTP recovered more annotated nucleotide binders and other plausible protein groups, such as nucleic acid binders, and GTPase and kinase regulatory subunits, which have been observed to co-stabilize with catalytic subunits[6] (Fig. 5b). Overall, DLPTP recovered nucleotide binders with smaller maximal thermal stability fold changes in addition to the vast majority of the hits with high fold change found by a threshold-based approach (Supplementary Fig. 5b).

## Discussion

We present DLPTP, a method for the detection of proteins whose thermal stability is modulated by the presence of a ligand from 2D-TPP data. Our approach builds upon the method of Storey et al.[19] for the detection of time-variable genes from time-course microarray data and, in particular, it compares for each protein a null and an alternative smooth curve fit via an $F$-statistic. Additional features of our approach include the following: (i) empirical Bayes moderation of the $F$-statistics by sharing variance information across proteins[20]; (ii) use of a domain-specific parametric model: the alternative model is a sigmoidal dose–response curve based on biophysical and assay-specific knowledge that constrains certain parameters while allowing other variables to be fit flexibly. This more specialized model makes more parsimonious use of data than non-parametric curve smoothing such as used by Storey et al. and thus may be expected to provide better statistical performance. (iii) To achieve FDR control, we adapted the bootstrap approach of Storey et al.[19] to the specific noise and replication structure of 2D-TPP experiments, namely, we restricted resampling to data obtained from the same MS run and introduced stratification of the set of proteins by number of measurements.

Our approach has the advantage of not relying on bespoke thresholds that have no clear performance characteristics (specificity and sensitivity) and are difficult to choose objectively across datasets with potentially different levels of noise and signal. The detection threshold of DLPTP is measured in terms of the FDR, which is an intuitive quantity that is comparable across experiments. We show that DLPTP indeed controls FDR by applying it to a synthetic dataset. However, for the moderated version of DLPTP, a conservative FDR estimation was observed, which may, in part, be due to the finite nature of the dataset. Yet,

DLPTP including moderation showed the best sensitivity-specificity tradeoff of all methods we compared.

We demonstrate the method's performance on primary data by applying it to previously published[2,23] and novel 2D-TPP datasets. We show that DLPTP is more sensitive than a previously proposed threshold-based approach and finds cognate targets and off-targets of multiple drugs and a metabolite. Application of DLPTP to 2D-TPP datasets profiling the HDAC8 inhibitor PCI-34051 and its analog BRD-3811 let us discover that both compounds inhibit LAP3, an interesting ovarian and liver cancer target. This opens the possibility of developing specific LAP3 inhibitors on the basis of BRD-3811.

Similar to any screening method, DLPTP may not detect all interactors of a ligand, i.e., allow false negatives. For instance, in the analysis of the panobinostat dataset, it missed HDAC6 at our chosen FDR. This misdetection is due to a small number of noisy measurements in the thermal profile of this protein, which prevented the alternative model from obtaining low residual errors, in spite of a visible dose–response trend (Supplementary Fig. 3e, especially at 51.9 °C). In general, such situations may arise for proteins quantified by only a small number of peptides. In the future, such problems are expected to be diminished with new MS instruments that provide greater protein coverage depth and, with the use of TMTpro[27], that allows multiplexing eight different ligand dosages at each temperature, while maintaining the same throughput.

In conclusion, we hope that the presented computational method will deliver high sensitivity for detecting ligand–protein interactions from TPP experiments with drugs and other ligands at low FDR, and will help make analyses of different datasets more comparable and more objective.

## Methods

**2D-TPP experiments**. 2D-TPP experiments for profiling PCI-34051 and BRD-3811 were performed as described[2]: HL-60 (DSMZ, ACC-3) cells were grown in Iscove's modified Dulbecco's medium supplied with 10% fetal bovine serum (FBS). Cells were treated with a concentration range (0, 0.04, 0.29, 2, 10 μM) of PCI-34051 (Selleckchem) or BRD-3811 (synthesized in-house[28] with >99% purity as determined by HPLC-UV254 nm) for 90 min at 37 °C, 5% CO$_2$. The samples from each treatment concentration were split into 12 portions, which were then heated each at a different temperature (42–63.9 °C) for 3 min and then incubated at room temperature for 3 min. Next, 30 μl of ice-cold phosphate-buffered saline (PBS) (2.67 mM KCl, 1.5 mM KH$_2$PO$_4$, 137 mM NaCl, and 8.1 mM NaH$_2$PO$_4$ pH 7.4) were supplemented with 0.67% NP-40 and protease inhibitors were added to the samples. Subsequently, cells were frozen in liquid nitrogen for 1 min, briefly thawed in a metal block at 25 °C, and then placed on ice and resuspended by pipetting. Samples were then incubated with benzonase for 1 h at 4 °C, followed by centrifugation at 100,000 × g for 20 min at 4 °C. Then, 30 μl supernatant were transferred into a new tube and were subjected to gel electrophoresis and sample preparation for MS analysis.

The 2D-TPP experiment to assess GTP-binding proteins was performed using gel-filtered lysate as described[2]. In short, Jurkat E6.1 cells (ATCC, TIB-152) were cultured in RPMI (GIBCO) medium supplemented with 10% heat-inactivated FBS. The cells were collected and washed with PBS. The cell pellet was resuspended in lysis buffer (PBS containing protease inhibitors and 1.5 mM MgCl$_2$) equal to ten times the volume of the cell pellet. The cell suspension was lysed by mechanical disruption using a Dounce homogenizer (20 strokes) and treated with benzonase (25 U/ml) for 60 min at 4 °C on a shaking platform. The lysate was ultracentrifuged at 100,000 × g, 4 °C for 30 min. The supernatant was collected and desalted using PD-10 column (GE Healthcare). The protein concentration of the eluted lysate was measured using Bradford assay. The protein concentration of the lysate was maintained at 2 mg/ml for the assay. The lysate was treated using a concentration range of GTP (0, 0.001, 0.01, 0.1, 0.5 mM) for 10 min at room temperature. The samples from each GTP concentration were split into 12 portions, which were then heated each at a temperature (42–63.9 °C) for 3 min. Post-heat treatment, the protein aggregates were removed using ultracentrifugation at 100,000 × g, 4 °C for 20 min. Subsequently, the supernatants were processed as described above.

**Protein identification and quantification**. Raw MS data were processed with Isobarquant[11] and searched with Mascot 2.4 (Matrix Science) against the human proteome (FASTA file downloaded from Uniprot, ProteomeID: UP000005640)

extended by known contaminants and reversed protein sequences (search parameters: trypsin; missed cleavages 3; peptide tolerance 10 p.p.m.; MS/MS tolerance 0.02 Da; fixed modifications were carbamidomethyl on cysteines and TMT10-plex on lysine; variable modifications included acetylation on protein N terminus, oxidation of methionine, and TMT10-plex on peptide N termini). Protein FDR was calculated using the picked approach[29].

Reporter ion spectra of unique peptides were summarized to the protein level to obtain the quantification $s_{i,u}$ for protein $i$ measured in condition $u = (j, k)$, i.e., at temperature $j$ and concentration $k$. Isobarquant additionally computes robust estimates of fold change $r_{i,u}$ for each protein $i$ in condition $u$ relative to control condition $u'$ using a bootstrap approach. We used these to obtain per-condition log2 signal intensities computed as $y_{i,u} = \log_2((r_{i,u}/\sum_l r_{i,l})\sum_l s_{i,l})$. This is one particular choice of protein quantification; we expect that our method can be used equivalently with input from other quantification methods.

In the resulting abundance table $Y = (y_{i,u})$, entries for which the value $r_{i,u}$ was obtained by not more than one peptide were marked as unreliable (i.e., set to not available (NA) in the software). For each protein $i$ we computed the total number of non-NA measurements $p_i$ and only retained proteins with $p_i \geq 20$ for subsequent analysis. In other words, proteins had to be quantified at least at four different temperatures and five different ligand concentrations each to be included in our analysis.

The MS experiment comprising the temperatures 54 and 56.1 °C was excluded from the analysis of the PCI-34051 dataset as we noticed that it contained unexpectedly high noise levels. In particular the relative reporter ion intensities at 54 °C showed about ten times higher variance than all other temperatures, likely due to a drop in instrument performance during the time this sample was measured.

Moreover, in the PCI-34051 and BRD-3811 datasets, we noted that measured profiles of some proteins appeared to have been affected by carry-over from previous experiments. These profiles exhibited a characteristic pattern as depicted in Supplementary Fig. 4c in which apparent stabilization of these proteins was observed only in half of the TMT channels corresponding to every other temperature. These proteins were filtered out by manual inspection.

**Data pre-processing of public datasets**. The panobinostat and JQ1 datasets were downloaded from the publisher websites as spreadsheets provided as supplementary data together with the publications[2,23]. Abundance tables $Y$ were computed and filtered as described above.

**Model description**. Two nested models were fitted to the abundance values of each protein $i$ at temperature $j$ and ligand concentration $k$. The null model is:

$$y_{i,j,k} = \beta_{i,j}^{(0)} + \epsilon_{i,j,k}^{(0)}. \tag{1}$$

Here, the base intensity level at temperature $j$ is $\beta_{i,j}^{(0)}$, and $\epsilon_{i,j,k}^{(0)}$ is a residual noise term. The alternative model is:

$$y_{i,j,k} = \beta_{i,j}^{(1)} + \frac{\alpha_{i,j}\delta_i}{1 + \exp(-\kappa_i(c_k - \zeta_i(T_j)))} + \epsilon_{i,j,k}^{(1)}. \tag{2}$$

Here, $\beta_{i,j}^{(1)}$ is again the base intensity level at temperature $j$, the parameter $\delta_i$ describes the maximal absolute stabilization across the temperature range, $\alpha_{i,j} \in [0, 1]$ indicates how much of the maximal stabilization occurs at temperature $j$ and $\kappa_i$ is a common slope factor fitted across all temperatures. Finally, $\zeta_i(T_j)$ is the concentration of the half-maximal stabilization (i.e., pEC50), with $\zeta_i(T_j) = \zeta_i^0 + a_i T$, where $a_i$ is a slope representing a linear temperature-dependent decay or increase of the inflection point, and $\zeta_i^0$ is the intercept of the linear model. Again, $\epsilon_{i,j,k}^{(1)}$ is a residual noise term. Both models were fit by minimizing the sum of squared residuals $\mathrm{RSS}_i^{(0)} = \sum_j\sum_k(\epsilon_{i,j,k}^{(0)})^2$ and $\mathrm{RSS}_i^{(0)} = \sum_j\sum_k(\epsilon_{i,j,k}^{(1)})^2$ using the L-BFGS-B algorithm[30] through R's optim function.

The start values for the parameter $\beta_{i,j}^{(0)}$ and $\beta_{i,j}^{(1)}$ in the iterative fit of the respective models were initialized with the mean abundance $\bar{y}_{i,j}$ of protein $i$ at temperature $j$; $\alpha_{i,j}$ was initialized as $\alpha_{i,j} = 0$ for all $i$ and $j$; $\delta_i$ was set to the maximal difference between abundance values within a temperature for protein $i$; $\kappa_i$ was initialized as the slope estimated by a linear model across temperatures; $\zeta_i^0$ was set to the mean log$_{10}$ drug concentration used; and $a_i$ was set to 0. The two fitted models can be compared using the F-statistic:

$$F_i = \frac{\mathrm{RSS}_i^{(0)} - \mathrm{RSS}_i^{(1)}}{\mathrm{RSS}_i^{(1)}}\frac{d_2}{d_1}, \tag{3}$$

with the degrees of freedom $d_1 = v_1 - v_0$ and $d_2 = p_i - v_i$, where $p_i$ is the number of observations for protein $i$ that were fitted, and $v_0$ and $v_1$ are the number of parameters of the null and alternative model, respectively.

For inference, we used an *empirical Bayes moderated* version of (3), as implemented in the `squeezeVar` function in the R/Bioconductor package `limma`[31]. `squeezeVar` uses the observed variances $s_i^2 = \mathrm{RSS}_i^{(1)}/d_2$ to identify a

common value $s_0^2$ and shrinks each $s_i^2$ towards that value. The motivation for such moderation is to accept a small cost in increased bias for a large gain of increased precision. To do so, `squeezeVar` assumes that the true $\sigma_i^2$ are drawn from a scaled inverse $\chi^2$ distribution with parameter $s_0^2$:

$$\frac{1}{\sigma_i^2} \sim \frac{1}{d_0 s_0^2} \chi^2. \tag{4}$$

Using the assumption that the residuals follow a normal distribution, Bayes' theorem and the scaled inverse Chi-squared prior, it can be shown[20] that the expected value of the posterior of $\sigma_i^2$ given $s_i^2$ is

$$\tilde{s}_i^2 = \frac{d_0 s_0^2 + d_2 s_i^2}{d_0 + d_2}. \tag{5}$$

Here, the hyperparameters $s_0^2$ and $d_0$ are estimated by fitting a scaled $F$-distribution with $s_i^2 \sim s_0^2 F_{d_2, d_0}$. Details are described by Smyth et al.[20]. Thus, we computed moderated $F$-statistics with

$$\widetilde{F} = \frac{\text{RSS}_i^{(0)} - \text{RSS}_i^{(1)}}{\tilde{s}_i^2 d_1}. \tag{6}$$

**FDR estimation**. To estimate the FDR associated with a given threshold $\theta$ for the $F$-statistic obtained for a protein $i$ with $m_i n_i$ observations, we adapted the bootstrap approach of Storey et al.[19] as follows. To generate a null distribution the following was repeated $B$ times: (i) resample with replacement the residuals $\epsilon_{i,w}^1$ obtained from the alternative model fit for protein $i$ in MS experiment $w$ to obtain $\epsilon_{i,w}^{1*}$ and add them back to the corresponding fitted estimates of the null model to obtain $y_{i,w}^* = \mu_{i,t}^0 + \epsilon_{i,w}^{1*}$. (ii) Fit null and alternative models to $y_{i,w}^*$ and compute the moderated $F$-statistic $\widetilde{F}^{0b}$. An FDR was then computed by partitioning the set of proteins $\{1, \ldots, P\}$ into groups of proteins with similar number $D(p)$ of measurements, e.g., $\gamma(p) = \lfloor \frac{D(p)}{10} + \frac{1}{2} \rfloor$ and then

$$\widehat{\text{FDR}}_g(\theta) = \hat{\pi}_{0g}(\theta) \frac{\sum_{b=1}^{B} \#\{\widetilde{F}_p^{0b} \geq \theta \,|\, \gamma(p) = g\}}{B \cdot \#\{\widetilde{F}_p \geq \theta \,|\, \gamma(p) = g\}} \tag{7}$$

The proportion of true null events $\hat{\pi}_{0g}$ in the dataset of proteins in group $g$ was estimated by:

$$\hat{\pi}_{0g}(\theta) = \frac{B \cdot \#\{\widetilde{F}_p < \theta \,|\, \gamma(p) = g\}}{\sum_{b=1}^{B} \#\{\widetilde{F}_p^{0b} < \theta \,|\, \gamma(p) = g\}}. \tag{8}$$

In the case of the standard DLPTP approach, the same procedure as above was performed using non-moderated $F$-statistics.

**Fluorometric aminopeptidase assay**. LAP3 activity was determined using the Leucine Aminopeptidase Activity Assay Kit (Abcam, ab124627) and recombinant LAP3 (Origene, NM_015907). Recombinant LAP3 enzyme was dissolved in the kit assay buffer and incubated for 10 minutes at room temperature with vehicle (dimethyl sulfoxide) or 100 μM of either PCI-34051 or BRD-3811. All other assay steps were performed as described in the kit. Fluorescent signal (Ex/Em = 368/460 nm) was detected over 55 min.

**Reporting summary**. Further information on research design is available in the Nature Research Reporting Summary linked to this article.

## Data availability
All acquired mass spectrometry datasets (2D-TPP experiment of GTP treated gel-filtered Jurkat lysate, PCI-34051 and BRD-3811 treated HL-60 cells) were deposited on PRIDE (accession number: PXD016640). Re-analyzed datasets profiling Panobinostat and JQ1 were downloaded from the publishers' websites (https://doi.org/10.1038/nchembio.2185 and https://doi.org/10.1016/j.cell.2018.02.030). Supplementary Data 1–3 provide raw data used for analysis and interpretation. A reporting summary for this article is available as a Supplementary Information file. All other data supporting the findings of this study are available from the corresponding authors on reasonable request. Source data are provided with this paper.

## Code availability
The software is available free and open source as an R package from Bioconductor: https://bioconductor.org/packages/TPP2D. All code used to perform the analyses presented in this manuscript is available at: https://github.com/nkurzaw/TPP2D_analysis; https://doi.org/10.5281/zenodo.4061271.

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

## Acknowledgements
We thank Srishti Dar, Henrik Hammarén, Britta Velten, Carola Doce, Dorothee Childs, Toby Mathieson, Thilo Werner, Constantin Ahlmann-Eltze, and Stephan Gade for insightful discussions and critical feedback. This work was supported by the European

Molecular Biology Laboratory (EMBL). N.K. was supported by a fellowship of the EMBL International PhD program. S.A. is funded by the Deutsche Forschungsgemeinschaft, SFB 1036. W.H. acknowledges funding from the European Commission's H2020 Programme, Collaborative research project SOUND (Grant Agreement number 633974).

## Author contributions

N.K., I.B., M.B., W.H., and M.M.S. conceived the project, designed experiments, and outlined desired method and software features. N.K. implemented and applied the software with input from S.A., W.H., and M.M.S., and performed data analysis. I.B. and S.S. performed experiments. H.F. and A.M. benchmarked and evaluated the method, and gave input. M.B., W.H., and M.M.S. jointly supervised the work. N.K. wrote the manuscript with feedback from all authors.

## Funding

## Competing interests

H.F., M.B., and M.M.S. are employees and/or shareholders of GlaxoSmithKline. The remaining authors declare no competing interests.
