## [Peer Review File · Nature Communications]

REVIEWER COMMENTS

Reviewer #1 (Remarks to the Author):

In their article, Kurzawa et al. propose a new statistical method to assess thermal stability as a function of ligand concentration in proteomic analyses.

The paper is well written and addresses an interesting interdisciplinary question at the crossing of biology, chemistry and statistics.

After a careful reading of the manuscript, I have the following major concerns:

- 1- Motivations need be completed,
- 2- Small amount of novelties,
- 3- Partial lack of methodological justifications,

which make me think that despite its interest and relevance, the article needs be revised. Please find below more detailed comments.

First, as is, the article gives the impression the proposed statistical method answers a question that is not necessarily the same as the practical one that motivated this work. This may come from a lack of sufficient explanation, so that more details would correct for this; or that the authors fell into the p-value bias, which consists in supposing that any "solid theoretical justification" should take the form of a statistical test (followed by multiple corrections). I do not know. However, my feeling comes from the motivating critical analysis of Becher et al.:

"... non-linear dose-response curves were fitted to each protein for each individual temperature. Subsequently, hits were defined by applying bespoke rules, including a requirement for two dose-response curves at consecutive temperatures to both have $R^2 > 0.8$ and a fold change of at least 1.5 at the highest treatment concentration. However, this approach, with its reliance on data-independent thresholds, has uncontrolled specificity [...], and as a consequence, may have suboptimal sensitivity if, e.g., the thresholds are too stringent."

Fitting a dose-response curve and then applying thresholds (either manually tuned or trained on data) to make a decision can be scientifically supported, depending on the biological/chemical questions at stake. It simply amounts to modeling the data and then categorizing them with more or less educated methods, and to focus on a data subset. On the other hand, the null hypothesis-testing framework requires defining an H_0 which corresponds to a "normal behavior" as opposed to the alternative "outlying behavior" of interest; and having Type I errors that are much more critical than Type II errors, so that a procedure to control the former ones despite looser

control on the latter ones is justified. In absence of such framework, there is no reason to define or use a statistical test, except possibly to easily give theoretical credits to biology-oriented readers or reviewers. Finally, the authors must contextualize and justify their choice to rely on null hypothesis significance testing.

My second observation concerns the amount of novelties in the article. The authors:

- elaborated on the question of a pre-existing article (Bosher et al.),
- relied on several pre-existing datasets to perform benchmarking and some re-analysis,
- performed few additional wet-lab experiments which do not carry any novelty (to the best of my understanding, as it is not my field of expertise)
- proposed a new statistical method, which according to the abstract, is essentially where the novelty of the article lies.

However, the statistical method is not as novel as claimed. According to the authors, their method consists is (1) a new statistical test; (2) Storey bootstrap method to control the FDR (ref. [15] in the article). However, this is not entirely true: First, the statistical test methodology is mainstream (it is described here in Wikipedia: https://en.wikipedia.org/wiki/F-test#Regression_problems). Second, using it to compare trends in various omic assays is not new, as this is precisely what is proposed in the same Storey article as referred to for the bootstrap method (ie ref. [15]). Finally, the main novelty is the replacement of the H_1 trend (eq.2) by a sigmoid function. Thus, if I had to reformulate a bit abruptly the novelty of the article, I think the authors should have written: "As statistical method, we propose to reuse that of [15], i.e. a nested hypothesis F-test followed by an FDR estimate relying on bootstrapping for better empirical null estimation. However, we modified the alternative hypothesis of the test according to the specific curves observed in dose responses, which differs from the microarray trends of [15]". This formulation would be less rotund but more honest. Despite smaller methodological inputs, the authors' proposal remain interesting (application oriented interest, R package, data reanalysis), but considering, publication in a journal as selective as Nature Communication may not be appropriate. Personally, I feel it would be more adapted to a technical note in a proteomics journal.

My third observation concerns some insufficient methodological justifications:

- First, in the nested model test, it is essential that the two models are nested. Based on equations 1 and 2, this is obviously the case, at least from a theoretical viewpoint. However, if in practice the estimation method at use cannot fit the parameters so that the sigmoid term is boiled down, then, the models are not nested anymore. The authors should provide guarantees with this respect (this can take the form of few lines depicting the parameter estimation method).
- Second, According to Suppl. Note Figure 1, and contrarily to what the authors claim ("its sensitivity was higher than the threshold-based approach") the specificity/sensitivity trade-off is in favor of the threshold-based approach. In my view, it is not inherently a problem, as comparison with this

method is more an approach issue than a performance issue (see point 1). Thus, I advise the authors to replace this figure by an estimate vs. ground true scatter plot, where the reader can easily check the conservativeness of the approach by checking the positions of the points with respect to the diagonal line.

- Third, According to Suppl. Note Figure 1, the proposed method may be anticonservative for a too low number of iterations B . This essential issue cannot appear in the Supp Mat only. It should be discussed in the article to avoid practitioners tune B to a too low value, as it would increase the putative H_1 list at a given FDR threshold.

To conclude, the article is interesting and proposes to elaborate on pre-existing tools to improve the analysis of thermal stability as a function of ligand concentration in proteomic analyses. However, the authors largely reused a pre-existing methodology without sufficiently crediting the original work, even though I acknowledge some novelties (see point 2 above). The authors did so without sufficiently justifying this type of methodology was suited to their problem (point 1) and did not sufficiently support the few modifications they proposed to the original method (see point 3). However, in my opinion, these presentation issues could be easily dealt with. In addition, the re-analysis of pre-existing datasets is interesting and the authors proposes a companion R package, making me think that, in the end, this work should deserve publication (even though a technical brief format in a more targeted journal would be probably more appropriated).

Reviewer #2 (Remarks to the Author):

Thermal Proteome Profiling (TPP) is a robust tool of identifying drug or metabolite targets. However, TPP data analysis has always been complicated since curves are being compared. The complexity of various protein behaviors in TPP experiment also adds to the difficulties. It's even more intricate when a second dimension (ligand concentration) is added besides temperature in 2-D TPP. Previous method used hard thresholds to identify targets without a FDR control. Kurzawa et al. presented a statistical analysis method with reliable control of the FDR for 2D TPP experiment, and a cognate open-source R package for it. The method is well explained and presented. It will be a significant tool in this field. However, there are still a few areas that could use elaboration/clarification to improve the manuscript.

1. The abstract could use some improvement if length is not a problem. For example, use one sentence to summarize what the major issue of previous statistical analysis method, and be specific that the new method is for two dimensional TPP or CETSA experiment, since there are more than one type of experiments for protein thermal stability profiling.

2. Fig. S2 and Fig. S3e, y-axis labels should be "ion intensity", not "ion area" since they are TMT data.

3. It will be much easier to read figures like Fig. S2 if dots are in a different color from the line color (or change the dot shape).

4. Fig. 1b, Fig. 1c., it's not very straightforward for a person not in statistics field to understand what asinh (F statistics) or RSS means. Showing the original 2D-TPP data (e.g., the small green heatmap used in the GTP/ATP paper in Nature Communications in 2019 by Savitski group) below these two panels for a couple of representative proteins can make it more understandable. For example, show what the original data look like for proteins uniquely identified in the new method, and show two proteins one of high F statistic and high $\log_2(\text{delta RSS})$, the other of relatively low F statistic and low $\log_2(\text{delta RSS})$, or proteins showing non-sigmoidal curves.

5. It seems only proteins of $P_i \geq 20$ observations are kept for subsequent analysis. For TMT 11-plex, does this mean 2 plexes (2 concentrations)? It makes sense that more targets can be identified if there are fewer missing values (Fig. S3 and S5). Some discussion on this (and be more specific what 20 observations means) can help readers understand it.

Reviewer #1 (Remarks to the Author):

In their article, Kurzawa et al. propose a new statistical method to assess thermal stability as a function of ligand concentration in proteomic analyses.

The paper is well written and addresses an interesting interdisciplinary question at the crossing of biology, chemistry and statistics.

We thank the reviewer for appreciating our work and their time and effort spent to review our manuscript. Based on their comments we have made several changes to the manuscript. Most prominently, we have improved our method by using an empirical Bayes approach together with the previously suggested test statistic. This leads to a significant improvement of the method's specificity-sensitivity tradeoff, which is now generally superior to the threshold-based approach of Becher et al. In addition, it has the benefit of giving the user control over the false-discovery rate at any chosen level.

After a careful reading of the manuscript, I have the following major concerns:

- 1- Motivations need be completed,
- 2- Small amount of novelties,
- 3- Partial lack of methodological justifications,

which make me think that despite its interest and relevance, the article needs be revised. Please find below more detailed comments.

We thank the reviewer for raising these fundamental and central points. The revision addresses each of them: the method has now a higher level of innovation, is methodologically better justified, and its applications are better motivated. We believe that there is now an even better argument for publication of this work.

First, as is, the article gives the impression the proposed statistical method answers a question that is not necessarily the same as the practical one that motivated this work. This may come from a lack of sufficient explanation, so that more details would correct for this; or that the authors felt into the p-value bias, which consists in supposing that any "solid theoretical justification" should take the form of a statistical test (followed by multiple corrections). I do not know. However, my feeling comes from the motivating critical analysis of Becher et al.:

"... non-linear dose-response curves were fitted to each protein for each individual temperature. Subsequently, hits were defined by applying bespoke rules, including a requirement for two dose-response curves at consecutive temperatures to both have $R^2 > 0.8$ and a fold change of at least 1.5 at the highest treatment concentration. However, this approach, with its reliance on data-independent thresholds, has uncontrolled specificity [...], and as a consequence, may have suboptimal sensitivity if, e.g., the thresholds are too stringent."

Fitting a dose-response curve and then applying thresholds (either manually tuned or trained on data) to make a decision can be scientifically supported, depending on the biological/chemical questions at stake. It simply amounts to modeling the data and then categorizing them with more or less educated methods, and to focus on a data subset.

We agree with the reviewer that 'bespoke-thresholds' approaches can be scientifically valid, and we have used such approaches in previous work. However, an approach based on statistical testing theory

has two important advantages over the bespoke-thresholds approach: first, it is less inviting to post-data-acquisition tinkering with thresholds until a desired result is reached. While such “hypothesis hacking” can probably not be totally eliminated, the statistical testing framework is supported by well-established theory, uses fewer parameters that can be tuned, is conservative by putting direct focus on type-I error (false discoveries) and provides better comparability of the thresholded quantities (p-values, FDRs) across experiments. Second, it provides better scalability and possibility of automation of the analysis; this is particularly important for future uses of the 2D-TPP technology, when the aim may be to analyse not just a handful of experiments, but hundreds or thousands—which might soon become feasible with the rapid improvement in mass spectrometry acquisition speed.

A second innovation of our approach regards the fitting of the dose-response curves. With “naive” fitting such as used previously for 2D-TPP datasets, concentration-dependent curves need to be fitted to as few as 5 data points. This can, and does, lead to overfitting. Our approach, which constrains certain curve parameters across temperatures, decreases the number of free parameters and thus leads to more stable estimates. This, in turn, leads to more reliable and more stable interpretation of the data. These points are discussed in the revised manuscript.

On the other hand, the null hypothesis-testing framework requires defining an H naught which corresponds to a "normal behavior" as opposed to the alternative "outlying behavior" of interest; and having Type I errors that are much more critical than Type II errors, so that a procedure to control the former ones despite looser control on the latter ones is justified. In absence of such framework, there is no reason to define or use a statistical test, except possibly to easily give theoretical credits to biology-oriented readers or reviewers. Finally, the authors must contextualize and justify their choice to rely on null hypothesis significance testing.

We agree with the reviewer. We note that low type I error is equivalent to high specificity, and low type II error to high sensitivity, so these concepts are related and generally important with any approach, regardless whether based on hypothesis testing theory or something else. Focus on type I error and false discovery rates is justified and appropriate in screening settings, such as the one here, where the aim is to screen the whole (technologically detectable) proteome for potential binders. A suitable number of “hits” are then typically followed up by more in-depth, more expensive analyses. It is an elegant and practically convenient property of the testing framework that it is possible to control type I error relatively easily, based on the mathematical and/or computational tractability of the null hypothesis. As for power/ type II error/ sensitivity: its assessment within the testing framework is no more and no less involved as in any other setting (e.g., in the bespoke thresholds framework), and relies on the availability of suitable positive control data based on which the sensitivity can be estimated. Thus, in a nutshell, the choice of the hypothesis testing approach gives us a convenient, intuitive and comparable-across-experiments type-I error control, and has no downsides compared to any other possible approach regarding type-II error. Moreover we note that our approach did detect more plausible binders (i.e., had better power / smaller type II error) than the threshold-based approach on the same data.

Moreover, the choice of null hypothesis or null model in our case is natural, and consistent with basic physics/chemistry: it allows arbitrary dependence of non-denatured protein intensity on temperature, but assumes independence on the concentration of the drug (potential ligand).

My second observation concerns the amount of novelties in the article. The authors:

- elaborated on the question of a pre-existing article (Bosher et al.),
- relied on several pre-existing datasets to perform benchmarking and some re-analysis,
- performed few additional wet-lab experiments which do not carry any novelty (to the best of my understanding, as it is not my field of expertise)
- proposed a new statistical method, which according to the abstract, is essentially where the novelty of the article lies.

Indeed, the main novelty here is the new analytical (statistical) method. However, for this it is not sufficient to just propose a method - it must be evaluated on real, relevant datasets and compared to previous and/or competing approaches. To this end, it is a good thing if at least some of these datasets have been previously published and analysed—it shows that the method is generally applicable and not just fine-tuned for one idiosyncratic dataset. Nevertheless, we have generated two new drug 2D-TPP datasets of yet unprofiled epigenetic drugs (PCI-3405 and BRD-3811) in HL60 cells. We find LAP3, a protein whose knockdown has been described to suppress ovarian cancer invasion (Wang et al., 2015), as a hitherto undescribed off-target of PCI-3405 and BRD-3811. We additionally validate LAP3 inhibition by both compounds in an in vitro assay. This opens the possibility for the design of specific LAP3 inhibitors in the future which may be interesting drug candidates.

Additionally, we generated a 2D-TPP dataset on Guanosine triphosphate (GTP) to showcase the applicability of our method to datasets profiling metabolite-protein interactions.

We acknowledge that this aspect of our work—the production of the three new 2D-TPP datasets—was not made sufficiently clear in the previous manuscript, therefore, we have now added an additional sentence to the abstract to point this out. Moreover, we have moved the paragraph describing the analysis of the GTP dataset from the Supplementary Note to the main text.

However, the statistical method is not as novel as claimed. According to the authors, their method consists is (1) a new statistical test; (2) Storey bootstrap method to control the FDR (ref. [15] in the article). However, this is not entirely true: First, the statistical test methodology is mainstream (it is described here in Wikipedia: https://en.wikipedia.org/wiki/F-test#Regression_problems). Second, using it to compare trends in various omic assays is not new, as this is precisely what is proposed in the same Storey article as referred to for the bootstrap method (ie ref. [15]). Finally, the main novelty is the replacement of the H_1 trend (eq.2) by a sigmoid function. Thus, if I had to reformulate a bit abruptly the novelty of the article, I think the authors should have written: "As statistical method, we propose to reuse that of [15], i.e. a nested hypothesis F-test followed by an FDR estimate relying on bootstrapping for better empirical null estimation. However, we modified the alternative hypothesis of the test according to the specific curves observed in dose responses, which differs from the microarray trends of [15]". This formulation would be less rotund but more honest.

We apologize for the lack of clarity, since we do not want to claim to have invented a completely new statistical test. Instead, we have adapted and suitably extended the work of Storey et al. to ligand dose range thermal proteome profiling (2D-TPP) experiments. The main novelty and research contribution is not the basic conceptual idea (which might be quite obvious to anyone trained in the field) but the demonstration that and how it actually works on real datasets, and that it has favourable performance characteristics. To clarify this, we have inserted a sentence in the manuscript: "We developed an approach that fits two nested models to protein abundances obtained from 2D-TPP. Our approach adapts and extends a method by Storey et al. (2005) for the analysis of microarray time course experiments¹⁸."

Our approach constitutes more than a simple application of the approach by Storey et al. to 2D-TPP data. As the reviewer acknowledges, our approach encompasses (a) problem-specific choice of null and alternative models, (b) adapting the resampling scheme to account for the specific noise structure of 2D-TPP experiments, and (c) expanding the empirical FDR estimation to the irregular replication characteristics of 2D-TPP, by using a group-specific approach (stratified by different numbers of observations per protein). In particular, the design of the alternative model represents a very different approach compared to the spline fit by Storey et al. and leverages relevant domain knowledge to choose appropriate constraints across temperatures to achieve a desirable trade-off of the model complexity between flexibility and data limitations. Further, as acknowledged by the reviewer, we supply a user-friendly software implementation which enables experimentalists to use our approach to analyse their data—something that is not possible by using the ‘edge’ package supplied by Storey and co-workers. We discuss these points in the updated manuscript.

Despite smaller methodological inputs, the authors' proposal remain interesting (application oriented interest, R package, data reanalysis), but considering, publication in a journal as selective as Nature Communication may not be appropriate. Personally, I feel it would be more adapted to a technical note in a proteomics journal.

We believe that given the above arguments, this work is of interest to the broad readership of Nature Communications. In particular, we do not “only” introduce a new analytical method, but we also use it to find new inhibitors of LAP3, a potentially interesting ovarian cancer and hepatocellular carcinoma target.

My third observation concerns some insufficient methodological justifications:

- First, in the nested model test, it is essential that the two models are nested. Based on equations 1 and 2, this is obviously the case, at least from a theoretical viewpoint. However, if in practice the estimation method at use cannot fit the parameters so that the sigmoid term is boiled down, then, the models are not nested anymore. The authors should provide guarantees with this respect (this can take the form of few lines depicting the parameter estimation method).

The optimization of the error functions for the alternative and null models is performed using the L-BFGS-B algorithm through R's *optim* function. We have now added a sentence to our methods section stating this. For the optimization for the fit of the alternative model (2), the starting parameters $\alpha_{i,j}$ are initialized with 0 for all i, j , and the $\beta_{i,j}$ with the abundance average at temperature j . This amounts to no (de-)stabilization and reduces the model to the null model (1). If these starting values are already the optimal fit to the data, then the algorithm will return them and not produce different values. As the reviewer also noted, the two models are nested, and this nestedness is a property of these models, irrespective of the parameter fit. Thus the models are guaranteed to be nested.

- Second, According to Suppl. Note Figure 1, and contrarily to what the authors claim ("its sensitivity was higher than the threshold-based approach") the specificity/sensitivity trade-off is in favor of the threshold-based approach. In my view, it is not inherently a problem, as comparison with this method is more an approach issue than a performance issue (see point 1).

Indeed, our approach in the first submission did not have, on the data shown, a better specificity/sensitivity trade-off compared to the threshold-based approach used by Becher et al. Even so, as the reviewer notes, it was conceptually more attractive by providing explicit control over the false-discovery rate, something that is not possible with the threshold-based approach.

However, in the revised version of the method, it *does* now have a better specificity/sensitivity. We extended and improved our method by making use of an empirical Bayes approach to moderate the *F*-statistics. The new method still provides all the same benefits as the previous version. In addition, the empirical Bayes moderation approach has been shown in numerous applications in high-throughput biology (e.g. microarray and RNA-Seq analysis (Ritchie et al. 2015; Robinson et al. 2010; Love et al. 2014)) to provide superior performance by trading a small increase in bias for a large reduction in stochastic variability (in other words, by being less susceptible to statistical chance). Indeed, it is also superior on the benchmark data to the threshold-based approach of Becher et al. in terms of the specificity/sensitivity trade-off.

Thus, I advise the authors to replace this figure by an estimate vs. ground true scatter plot, where the reader can easily check the conservativeness of the approach by checking the positions of the points with respect to the diagonal line.

As the reviewer suggested, we have replaced the old benchmark figure with a two panel figure: one showing the estimate vs. ground truth plot and one showing a reduced version of our previous FDR-TPR plot. This figure is now included in the results part of the manuscript.

- Third, According to Suppl. Note Figure 1, the proposed method may be anticonservative for a too low number of iterations *B*. This essential issue cannot appear in the Supp Mat only. It should be discussed in the article to avoid practitioners tune *B* to a too low value, as it would increase the putative *H*₁ list at a given FDR threshold.

We agree with the reviewer and we now only consider $B = 100$ bootstraps, to not give the wrong impression that as few as $B = 5$ should be done. Furthermore, we described this already previously in the package vignette (step-by-step analysis manual) advising users to use a minimum of $B = 20$ rounds of bootstrapping. Additionally, the software now emits a warning when a user calls the bootstrapping function with $B < 20$ in the latest version of our package.

To conclude, the article is interesting and proposes to elaborate on pre-existing tools to improve the analysis of thermal stability as a function of ligand concentration in proteomic analyses. However, the authors largely reused a pre-existing methodology without sufficiently crediting the original work, even though I acknowledge some novelties (see point 2 above). The authors did so without sufficiently justifying this type of methodology was suited to their problem (point 1) and did not sufficiently support the few modifications they proposed to the original method (see point 3). However, in my opinion, these presentation issues could be easily dealt with. In addition, the re-analysis of pre-existing datasets is interesting and the authors proposes a companion R package, making me think that, in the end, this work should deserve publication (even though a technical brief format in a more targeted journal would be probably more appropriated).

Reviewer #2 (Remarks to the Author):

Thermal Proteome Profiling (TPP) is a robust tool of identifying drug or metabolite targets. However, TPP data analysis has always been complicated since curves are being compared. The complexity of various protein behaviors in TPP experiment also adds to the difficulties. It's even more intricate when a second dimension (ligand concentration) is added besides temperature in 2-D TPP. Previous method used hard thresholds to identify targets without a FDR control. Kurzawa et al. presented a statistical analysis method with reliable control of the FDR for 2D TPP experiment, and a cognate open-source R package for it. The method is well explained and presented. It will be a significant tool in this field. However, there are still a few areas that could use elaboration/clarification to improve the manuscript.

We thank the reviewer for their positive comments on our work and their time and effort spent reviewing our manuscript. We have followed the reviewer's suggestions and substantially revised and extended the manuscript. Our point-by-point response to their comments follows:

1. The abstract could use some improvement if length is not a problem. For example, use one sentence to summarize what the major issue of previous statistical analysis method, and be specific that the new method is for two dimensional TPP or CETSA experiment, since there are more than one type of experiments for protein thermal stability profiling.

We have now extended the abstract to include more detailed information on the problems with previous analysis approaches and the exact TPP assay format this method targets.

2. Fig. S2 and Fig. S3e, y-axis labels should be "ion intensity", not "ion area" since they are TMT data.

This has been corrected as suggested.

3. It will be much easier to read figures like Fig. S2 if dots are in a different color from the line color (or change the dot shape).

This has been now adapted (lines are now dark gray, data points remain black).

4. Fig. 1b, Fig. 1c., it's not very straightforward for a person not in statistics field to understand what asinh (F statistics) or RSS means.

We agree that this was insufficiently explained previously. We have now replaced our previous transformation of the F -statistic using an inverse hyperbolic sine with a $\log_2(x + 1)$ -transformation which we believe will be more intuitive for less mathematically inclined readers. To better bring out stabilized as opposed to destabilized proteins we have also changed our previous volcano plots to two-sided ones. This is done by replacing the previous effect size measure on the x-axis $\log_2(\text{RSS}^0 - \text{RSS}^1)$ with an effect-directionality signed effect size using the square root of the differences between residual sum of squares between both models. Additionally, we have added an extensive explanation to the figure legend of Figure 1 to introduce these axes.

Showing the original 2D-TPP data (e.g., the small green heatmap used in the GTP/ATP paper in Nature Communications in 2019 by Savitski group) below these two panels for a couple of representative proteins can make it more understandable. For example, show what the original data look like for proteins uniquely identified in the new method, and show two proteins one of high F statistic and high $\log_2(\Delta \text{RSS})$, the other of relatively low F statistic and low $\log_2(\Delta \text{RSS})$, or proteins showing non-sigmoidal curves.

We have added additional supplementary visualizations of examples for various discussed targets with low and high F -statistics and $\log_2(\Delta \text{RSS})$.

5. It seems only proteins of $P_i \geq 20$ observations are kept for subsequent analysis. For TMT 11-plex, does this mean 2 plexes (2 concentrations)? It makes sense that more targets can be identified if there are fewer missing values (Fig. S3 and S5). Some discussion on this (and be more specific what 20 observations means) can help readers understand it.

We have now added an additional sentence clarifying this: “For each protein i we computed the total number of non-NA measurements p_i and only retained proteins with $p_i \geq 20$ for subsequent analysis. In other words, proteins had to be quantified at least at four different temperatures and five different ligand concentrations each to be included in our analysis.”

REVIEWERS' COMMENTS

Reviewer #1 (Remarks to the Author):

In this revised version, the authors largely answered my questions and improved the article, but also the underlying tool. While I was initially skeptical with the amount of novelties presented in the article, I am now convinced the article deserves publication.

However, I still have a major methodological issue that need be addressed: Figures 2a and 2b are somehow alarming, as clearly, the FDR control can be too liberal (i.e. anticonservative, as the computed FDR underestimate the real FDR), depending on the chosen method (with or without moderated statistics) and the expected FDR threshold (below or above 5%). Moreover, the difference between the observed and nominal FDRs (notably at low FDR) is such that the authors cannot state without further explanation that their FDR is well calibrated. As anti-conservative FDRs do not authorize FDR control, a more refined analysis and a small discussion are necessary here, notably to report guidelines that will secure conservative (and calibrated) FDRs to the users.

Besides, here are a few other minor comments. Addressing them would likely lead to an improved manuscript:

- In the "Design of models for ligand dose range thermal profiles" subsection, the "inflection point (EC50)" is referred to but not explained
- In the "Model description" subsection, the added text (about the initialization) is difficult to read. Splitting the text with semi-column would help reading the concatenated sentences.
- Zooming in figures 2a and 2b to the FDR range of interest would be more relevant: no one is interested in controlling an FDR at 20%, or beyond. In my opinion, zooming in the [0; 10%] interval (for both figures) is important; possibly, a wider range (i.e. the zoomed out figures) can be kept in supplemental files.
- More details on how the moderated statistics was incorporated in the model would be insightful, as it is an interesting feature of the proposed method (the reader can so far only rely on a 3-line description quoting the R function at use).

Once these issues are addressed, the article can be accepted for publication.

Reviewer #1 (Remarks to the Author):

In this revised version, the authors largely answered my questions and improved the article, but also the underlying tool. While I was initially skeptical with the amount of novelties presented in the article, I am now convinced the article deserves publication.

However, I still have a major methodological issue that need be addressed: Figures 2a and 2b are somehow alarming, as clearly, the FDR control can be too liberal (i.e. anticonservative, as the computed FDR underestimate the real FDR), depending on the chosen method (with or without moderated statistics) and the expected FDR threshold (below or above 5%). Moreover, the difference between the observed and nominal FDRs (notably at low FDR) is such that the authors cannot state without further explanation that their FDR is well calibrated. As anti-conservative FDRs do not authorize FDR control, a more refined analysis and a small discussion are necessary here, notably to report guidelines that will secure conservative (and calibrated) FDRs to the users.

Thank you for pointing this out. In fact, we found a mistake in our code that was responsible for the anti-conservative behavior of the moderated version of our method. We have now fixed this and re-ran all analyses with the updated version. While the moderated version of DLPTP now appears conservative (which may in part be due to the finite nature of the simulated dataset, i.e. there is a limited number of protein profiles simulated under the null model that obtain high F -statistics and can be compared to the top true positives) it still shows the best sensitivity-specificity trade-off of all methods compared.

Besides, here are a few other minor comments. Addressing them would likely lead to an improved manuscript:

- In the "Design of models for ligand dose range thermal profiles" subsection, the "inflection point (EC50)" is referred to but not explained

A more detailed explanation of the EC50 has been added.

- In the "Model description" subsection, the added text (about the initialization) is difficult to read. Splitting the text with semi-column would help reading the concatenated sentences.

We have added semicolons to this section as suggested.

- Zooming in figures 2a and 2b to the FDR range of interest would be more relevant: no one is interested in controlling an FDR at 20%, or beyond. In my opinion, zooming

in the [0; 10%] interval (for both figures) is important; possibly, a wider range (i.e. the zoomed out figures) can be kept in supplemental files.

These figures are now represented more zoomed in as suggested.

- More details on how the moderated statistics was incorporated in the model would be insightful, as it is an interesting feature of the proposed method (the reader can so far only rely on a 3-line description quoting the R function at use).

This section has been extended to explain how the moderation of the F-statistic works formally.

Once these issues are addressed, the article can be accepted for publication.

We thank the reviewer for the time spent reviewing our work, valuable feedback and the recommendation of our manuscript for publication.